# Association of faecal pH with childhood stunting: Results from a cross-sectional study

Md. Shabab Hossain,[1] Subhasish Das,[1] Md. Amran Gazi,[1] Md. Ashraful Alam,[1] Nur Muhammad Shahedul Haque,[1] Mustafa Mahfuz,[1] Tahmeed Ahmed,[1] Chris J Damman[2,3]

[1]Nutrition and Clinical Services Division (NCSD), International Centre for Diarrhoeal Disease Research Bangladesh, Dhaka, Bangladesh
[2]Division of Gastroenterology, University of Washington, Seattle, Washington, USA
[3]Bill and Melinda Gates Foundation, Seattle, Washington, USA

**Correspondence to**
Dr Tahmeed Ahmed; tahmeed@icddrb.org

## ABSTRACT

**Background** Gut microbiota plays an important role in the growth of children. The gut of children with optimum growth is enriched in certain species, especially *Bifidobacteria* and *Clostridia*. *Bifidobacteria* and commensal *Clostridia* both contribute to formation of acidic stool, and an elevated faecal pH indicates reduction of these species in the gut. The purpose of the study was to investigate the association of faecal pH with childhood stunting.

**Methods** In this cross-sectional study, 100 children with length-for-age Z score (LAZ) <−1 aged between 12 and 18 months were enrolled from the ongoing Bangladesh Environmental Enteric Dysfunction study conducted in Dhaka, Bangladesh. LAZ was measured by anthropometry and data on factors affecting linear growth were recorded. Faecal pH measurement was done using pH metre on freshly collected non-diarrhoeal faecal samples following standard procedure. Multiple quantile regression was done to quantify the relation between faecal pH and LAZ scores.

**Results** The mean LAZ and faecal pH of the children were −2.12±0.80 and 5.84±1.11, respectively. Pearson correlation analysis showed a statistically significant negative correlation between stool pH and the LAZ scores (p<0.01). After inclusion of other factors affecting linear growth into the regression model, a statistically significant inverse association was observed between faecal pH and LAZ score (p<0.01).

**Conclusion** Elevated faecal pH was found to have a significant association with stunted growth. As an indicator of gut microbiota status, faecal pH might have emerged as a possible indirect determinant of childhood stunting.

**Trial registration number** NCT02812615

### What is known about the subject?

► Chronic malnutrition, manifested as linear growth failure or stunting, is the most common form of malnutrition across the globe.
► Alteration in gut microbiota has impact on stunted growth and microbiota of children with optimum growth are enriched in commensal *Clostridia* and *Bifidobacteria*.
► Abundances of commensal *Clostridia* and *Bifidobacteria* contribute to the formation of acidic stool and reductions of these species lead to an elevation in faecal pH.

### What this study adds?

► There is a significantly inverse association between faecal pH and child growth, making elevated faecal pH a possible indirect determinant of childhood stunting.

## INTRODUCTION

Malnutrition is the underlying cause of almost half of all deaths in five children, estimated to be more than 3 million child deaths annually.[1] The burden of malnutrition is very high in children living in low-income and middle-income countries (LMICs).[1] Chronic malnutrition, which is manifested as linear growth failure or stunting,[2] is the most common form of malnutrition across the globe, affecting about one-third of children in LMICs.[3] Among these LMICs, countries from the southern Asia, specifically Bangladesh, India, Pakistan, Nepal and Bhutan, harbour a substantially high number of stunted children.[4] Stunted children suffer from reduced neurodevelopmental potential and poor cognitive function,[2] thus fail to reach their full developmental potential,[5] leading to substandard educational performance and economic productivity in the later phases of life.[6] The aetiology of stunting is poorly understood[4] and though poverty-associated food insecurity might play a vital role, the aetiology of this condition is far more complex.[7] Several factors including intrauterine growth retardation, mother's height, impaired immunity and recurrent infections, small intestinal bacterial overgrowth (SIBO), environmental enteric dysfunction (EED), neuroendocrine and hormonal factors, and host genetic factors compounded by food insecurity are all part of this vicious process.[8–14] Recent studies

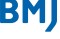

suggest that alteration in the gut microbiota is also a part of this process.[4 7–9 15]

A healthy gut microbiota is essential to human health.[7] It performs diverse protective, structural and metabolic functions of the gut; for instance, production of short-chain fatty acids (SCFAs), which act as the primary nutrient source for the colonic epithelium, intestinal epithelial cell proliferation and development of the mucosal immune system, production of certain vitamins, promoting mineral absorption, influencing metabolic rates and appetite, induction of host genes for nutrient uptake; all of which are critical for optimal nutrient absorption.[8 16–18] On the other hand, dysbiosis refers to an altered microbiota composition that could lead to a number of disease states or malnutrition even in the face of adequate food intake.[7]

Despite the fact that chronic malnutrition is more prevalent than acute malnutrition globally, most published studies have concentrated on the role of gut microbiota in acute malnutrition. The limited studies that exist show that, gut microbiota of stunted children is predominant in bacteria belonging to the inflammogenic taxa including those in the *Desulfovibrio* genus and *Campylobacterales* order.[4] On the contrary, the gut microbiota of children with optimum growth is enriched in certain species, especially *Bifidobacteria*,[4 7] which is referred to as the keystone infant gut symbiont[19] and the commensal *Clostridia*.[20] Faecal pH, a probable determinant whose relationship with growth has never been explored to the best of our knowledge, is strongly driven by the abundance of *Bifidobacteria* and the butyrate producing commensal *Clostridia*, especially clusters XIVa and IV, in the gut.[19 21] These commensals lower the pH in the colon contributing to formation of a more acidic stool and the reduction or loss of these species lead to an elevation in faecal pH.[19]

Therefore, elevated faecal pH indicates a marked change in the infant gut microbiome, specifically a profound reduction of the mentioned species. Since faecal pH is an indirect yet novel tool for assessing *Bifidobacteria* and commensal *Clostridia* abundance in the infant gut, while their abundances share a significant association with child growth, we made an attempt to determine the association between child growth and faecal pH and to investigate if elevated faecal pH is associated with stunted growth in early childhood.

## METHODS

### Study site and data collection

This cross-sectional study was a part of the ongoing Bangladesh Environmental Enteric Dysfunction (BEED) study, which is being conducted among the residents of Bauniabadh slum in Mirpur, Dhaka. BEED study is a community-based nutrition intervention study aimed to validate non-invasive biomarkers of EED and better understand the disease pathogenesis. The details and overall design of the BEED study have already been published

elsewhere.[22] A written informed consent was obtained from the parents of children after explaining the aims and procedures of the study. For this particular study, a total of 100 children aged between 12 and 18 months with length-for-age Z-scores (LAZ) <−1 were enrolled from the same community from September 2018 to February 2019. Weight and length of the child along with mother's height were measured using standard protocol.[23] Data regarding socioeconomic, morbidity, dietary intake and breastfeeding were collected using hard copy questionnaires at the time of enrolment and prior to sample collection.[24 25] Dietary intake was assessed using qualitative Food Frequency Questionnaires, in which individual food items that was fed to each child during the last 24 hours were recorded.[24 25]

### Measuring faecal pH

Prior to the nutritional intervention, the health workers waited for the children to defecate and faecal samples from the first bowel movement after enrolment were collected. Following standard operating procedures,[19 22 26] fresh non-diarrhoeal faecal samples were collected immediately after defecation into a plastic pot and 1–2 g from that stool sample was relocated in a sterile specimen container. Ten millilitres of deionised water was mixed and homogenised with this stool properly. The probes of the pH metre were completely dipped into this mixture and kept for 1 min and the reading was recorded. The procedure was done within the first 2 hours of sample collection. The test was carried out using the portable stool pH metre (Hanna Instruments, USA).

### Small intestinal bacterial overgrowth

Presence of SIBO was determined by the hydrogen breath test from breath samples taken from the children using BreathTracker SC (QuinTron Instrument Company, Milwaukee, USA). A total of 10 breath samples were collected per case including a 3-hour fasting baseline sample, followed by nine more samples at 20 min intervals after ingestion of glucose at 1 g/kg body weight diluted in 5 mL/kg of water. A single difference of ≥12 ppm of hydrogen value from fasting baseline was considered to be SIBO positive.[27]

### Statistical analyses

Statistical analyses were performed using Stata V.13.0. Mean values and SD of means were used to describe the distribution. Correlation between stool pH and LAZ scores was done by simple correlation analysis using Pearson correlation and visualised by scatter plots. As LAZ is the outcome variable, which was not normally distributed based on Shapiro-Wilk test, at first simple quantile regression was performed between LAZ scores with each individual factor. Results with a significance level at or below 0.2 were included in the multiple quantile regression model. In addition, age and sex were included in the final model due to the inconsistent growth pattern in children of either sex in this age group[28] and minimum

**Table 1** Demographic characteristics and results of the respondents

| Characteristics | Mean±SD (n=100) |
|---|---|
| Mean age of children (months) | 15.16±2.15 |
| Female, % | 57% |
| Faecal pH | 5.84±1.11 |
| Length for age Z-score | −2.12±0.80 |
| Weight for age Z-score | −2.18±6.9 |
| Weight for height Z-score | −0.73±0.9 |
| Mother's age (years) | 25.2±6.3 |
| Mother's schooling years | 5.96±2.8 |
| Mother's currently working status | |
| Yes | 25% |
| No | 75% |
| Father's occupation | |
| Labourer | 19% |
| Service holder/businessman | 22% |
| Driver | 28% |
| Factory worker | 17% |
| Others (unemployment, retired) | 14% |
| Mother's hand wash practice | |
| Wash hand with soap after helping child defecate, % | |
| Always | 75% |
| Sometimes | 22% |
| Rarely | 3% |
| Never | 0% |
| Wash hand with soap before preparing food, % | |
| Always | 15% |
| Sometimes | 36% |
| Rarely | 11% |
| Never | 38% |
| Monthly family income in USD, median (IQR) | 178 (119, 237) |

USD, United States Dollars.

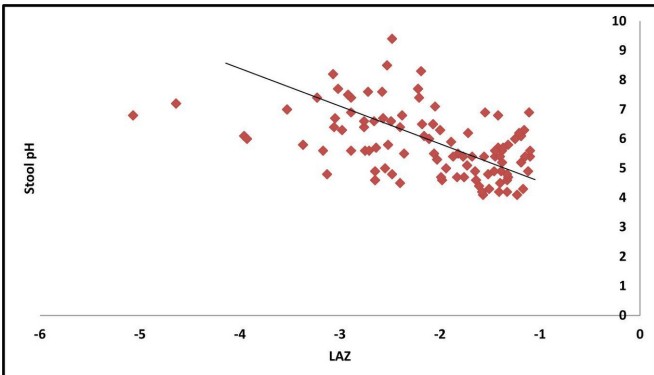

**Figure 1** Scatter diagram showing negative correlation between faecal pH and LAZ scores. LAZ, length-for-age Z score.

dietary diversity (MDD) was included in the final model instead of the individual food groups. Multiple quantile regression was done to quantify the relation between faecal pH and LAZ scores after adjusting for age, sex, maternal height, MDD, breast feeding, morbidity, antibiotics taken in the last 14 days, SIBO and incorporation of several aspects of socioeconomic status: access to improved water and sanitation, the selected approach to measuring household Wealth (Assets), Maternal education and Income index.[29]

## RESULTS

A total of 100 children with mean age of 15±2 months underwent faecal pH examination. Among them, 57 (57%) were female and 43 (43%) were male. Table 1 shows the sociodemographic characteristics of the study population.

The mean LAZ and faecal pH of the children were −2.12±0.80 and 5.84±1.11, respectively. Pearson correlation between faecal pH and LAZ scores showed a statistically significant negative correlation (−0.46, p<0.01). Scatter plot (figure 1) showed a similar correlation between stool pH and the LAZ scores as well.

Results of the simple quantile regression with a significance level at or below 0.2 and in addition age, sex and MDD were included in the multiple quantile regression model. SIBO was not significantly correlated with stunting in the simple quantile regression (p=0.14) and thus was not included in the multiple model. A statistically significant inverse association was observed between faecal pH with LAZ score of the children (table 2). A single unit increase in faecal pH was associated with 0.32 unit decrease in the LAZ score of the child.

## DISCUSSION

Our study revealed that elevated faecal pH has a significant association with stunted growth in early childhood. Though studies have shown significant association of faecal pH with severe acute malnutrition,[30] a thorough literature review yielded no results on the relationship between faecal pH with chronic malnutrition or stunted growth.

A trend of increasing infant faecal pH in children is observed over the past century, which demonstrated a change in faecal pH from 5.0 to 6.5 from 1926 to 2017.[19] This elevation of faecal pH is consistent with reported reduction of certain important gut microflora. Factor responsible for reduction of these important microbiota from the infant gut over the past century comprises an increase in consumption of human milk replacers (eg, infant formula), which lack the bacterial selectivity of human milk,[19] increased caesarean section delivery that limits the natural faecal-oral transfer from mother to infant associated with vaginal delivery[31] and overuse of antibiotics which alter the acquisition of gut microbes by the infant.[32] *Bifidobacteria* and butyrate producing

**Table 2** Association of LAZ scores with faecal pH

| Variables | Unadjusted | | | Adjusted | | |
|---|---|---|---|---|---|---|
| | β* | SE | P value * | β* | SE | P value * |
| Age (in months) | 0.03 | 0.06 | 0.65 | 0.03 | 0.04 | 0.44 |
| Sex | 0.16 | 0.26 | 0.55 | −0.02 | 0.18 | 0.93 |
| Mothers height (in cm) | 0.06 | 0.03 | 0.02 | 0.03 | 0.02 | 0.08 |
| Fever (in last 14 days) | 0.44 | 0.93 | 0.64 | | | |
| Antibiotics taken (in last 14 days) | 0.44 | 1.32 | 0.74 | | | |
| Faecal pH | −0.36 | 0.07 | <0.01 | −0.32 | 0.08 | <0.01 |
| SIBO | 0.57 | 0.38 | 0.14 | | | |
| MDD | 0.01 | 0.27 | 0.97 | 0.15 | 0.18 | 0.41 |
| Food group 1 (grain, roots and tubers) | −0.06 | 1.32 | 0.96 | | | |
| Food group 2 (legumes and nuts) | −0.35 | 0.24 | 0.15 | | | |
| Food group 3 (milk, yoghurt and cheese) | −0.13 | 0.26 | 0.62 | | | |
| Food group 4 (meat, fish and poultry) | 0.09 | 0.26 | 0.73 | | | |
| Food group 5 (eggs) | −0.05 | 0.26 | 0.85 | | | |
| Food group 6 (vitamin A rich fruits and vegetables) | −0.66 | 0.25 | 0.09 | | | |
| Food group 7 (other fruits and vegetables) | 0.30 | 0.25 | 0.23 | | | |
| Currently breast feeding | 0.09 | 0.33 | 0.79 | | | |
| WAMI | 0.29 | 1.02 | 0.78 | | | |

*β=Regression coefficient; p value=significance level.
LAZ, length-for-age Z score; MDD, minimum dietary diversity; SIBO, small intestinal bacterial overgrowth; WAMI, Water and sanitation, the selected approach to measuring household wealth (Assets), Maternal education and Income.

commensal *Clostridia* both contribute to the formation of acidic stool and an elevated faecal pH indicates a reduction of these species in the gut, which has been reported in many studies.[19 21 33]

While *Clostridium* cluster I and cluster XI include pathogenic species *Clostridium perfringens, Clostridium tetani* and *Clostridium difficile,* most of the *Clostridia* maintain a commensal relationship with the host.[21] Gut commensal *Clostridia* consist of gram-positive, rod-shaped bacteria and clusters XIVa and IV make up a substantial part (10%–40%) of the total bacteria in the gut microbiota.[21] These commensals perform their metabolic functions through the release of butyrate, which is an important SCFA, that lowers the pH in the colon and contributes in stool acidification.[21] 'Bacillus bifidus' or *Bifidobacterium* is a gram-positive, acidiphilic bacteria[34] belonging to the *Bifidobacteriaceae* family has the capability to consume human milk oligosaccharides (HMOs) and convert them into acidic end products with a meaningful effect on faecal pH.[19] So, faecal pH acts as a useful marker for assessing gut *Bifidobacterium* and commensal *Clostridia* status in non-diarrhoeal children. However, as HMOs are involved in the acidification process related to *Bifidobacterium,* the commensal *Clostridia* are likely to have an important role in children of the studied age group. However, using the data from the Bangladesh Demographic and Health Survey 2004, a study showed that the average duration of breast feeding in Bangladeshi children is 31.9 months.[35] So, children in this region are usually breastfed along with

complimentary feeding beyond 24 months of age, which supports the concept of presence of *Bifido* in the gut of children up to this age.[35] Characterisation of microbiome (SCFA producers) in the stool could not be investigated in the present study, which is the only way to clarify this topic. Further studies are warranted to make exploration in this regard.

In the developing world, most of the linear growth faltering occurs in the first 2 years of life. However, very few studies exist on gut microbiota of children belonging to this period of life when the maturation of microbiota is the most required.[7] As already mentioned, gut microbiota performs a wide range of protective, structural and metabolic functions of the gut, all of which are critical to optimal nutrient absorption.[8 16–18] And an optimal nutrient absorption as well as a well-developed immune system is the key to optimum growth. A recent study done in Guatemala has shown that both *Bifido* and *Clostridia* have biologically plausible effect on childhood stunting, however, the former was found to be more evident than the latter.[36] SIBO was not significantly correlated with stunting in the present study. This might be due to the fact that all the participants here were non-diarrhoeal, while few studies indicate that SIBO is associated with chronic diarrhoea.[37 38]

SCFAs, in particular butyrate, are calorie-rich nutrients and are the preferred energy source for colonocytes and also a potent regulator of host metabolism.[20] Their chronic depletion results in undernutrition and

stunting.[20] Furthermore, butyrate is an effector of colonisation resistance to facultative anaerobic enteropathogens and reduction in butyrate promotes over-representation of aerobic enteropathogenic taxa *Streptococcus, Neisseria, Staphylococcus, Haemophilus, Campylobacter* and *Escherichia/Shigella* genera constituting a bona fide dysbiosis, resulting in increased frequency and severity of gastrointestinal diseases, again leading to undernutrition and stunting.[20 39] Commensal *Clostridia* belonging to clusters XIVa and IV release butyrate to perform their metabolic functions and also been reported to be strong inducers of colonic T regulatory cell accumulation,[40] contributing to immune system development in children and reduction of this species is considered as a signature of stunting.[20] On the other hand, *Bifidobacterium* is called the keystone infant gut symbiont[19] and *Bifidobacterium longum* subsp. *infantis* (*B. infantis*) is unique among all gut microbes to have the capacity to consume the full range of HMOs.[41] HMOs serve as nutrients for colon bacteria, further promotes growth of non-pathogenic, bifidogenic microflora and prevents pathogenic microorganisms, for example, *Campylobacter jejuni, Escherichia coli, Vibrio cholera, Shigella* and *Salmonella* strains from adhesion to the host's epithelial surface and protect against infections and diarrhoea.[42]

This discussion makes it evident that *Bifidobacteria* and commensal *Clostridia* play a significant role for optimum child growth, both by facilitating nutrient absorption and providing with immunity, and studies show that these species are abundant in the gut of children with optimum growth and markedly decreased in the gut of stunted children.[4 7] And as faecal pH is an indirect yet novel measuring tool of their abundance in the non-diarrhoeal infant gut, which in turn is strongly associated with growth, elevated faecal pH might just have emerged as a possible indirect determinant of stunted growth in early childhood, as we have found in this study.

## Limitations

This study was conducted as a part of the BEED study and the target population of the parent study was only stunted (LAZ <−2) and at risk of stunting (LAZ <−1 to −2) children and therefore, by definition were not completely healthy. The pH results of healthy children could not be examined or compared. Characterisation of microbiome (SCFA producers) and SCFA profiles in the stool of the children with and without low pH was also not investigated.

## CONCLUSION

Elevated faecal pH was found to have significant association with stunted growth in early childhood. In this study, as an indicator of gut microbiota status, faecal pH came out as a possible indirect determinant of childhood stunting. Characterisation of microbiome (SCFA producers) in the stool could not be investigated in the present study, which is the only way to clarify this topic.

Further studies are warranted to make exploration in this regard.

**Acknowledgements** icddr,b acknowledges with gratitude the commitment of the BMGF to its research efforts. icddr,b is also grateful to the governments of Bangladesh, Canada, Sweden and the UK for providing core/unrestricted support. We express our sincere thanks to our colleagues at the International Centre for Diarrhoeal Disease Research, Bangladesh and the study participants.

**Contributors** CJD and TA originated the idea for the study and TA led the protocol design. MSH, SD and TA participated in the design of the study. MSH, SD, MAG, NMSH, MM and TA conducted the study and supervised the sample and data collection. MSH and MAG performed and supervised the sample analysis. MSH, SD, MAA, MM and TA were involved in the data analysis. MSH, SD and TA interpreted the results. MSH, SD, MM and TA were involved in the manuscript writing. All authors read and approved the final manuscript.

**Funding** This protocol is supported by the Bill and Melinda Gates Foundation under its Global Health Program. Project investment ID is OPP1136751. (https://www.gatesfoundation.org/How-We-Work/Quick-Links/GrantsDatabase/Grants/2015/11/OPP1136751).

**Competing interests** None declared.

**Patient consent for publication** Not required.

**Ethics approval** Ethical approval was taken from Ethical Review Committee (ERC) of the Institutional Review Board (IRB) of icddr,b.

**Provenance and peer review** Not commissioned; externally peer reviewed.

**Data availability statement** Data are available on reasonable request.

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
