## [Reviewer comments · BMJ Paediatrics Open]

ARTICLE DETAILS

TITLE (PROVISIONAL)	Association of fecal pH with childhood stunting: Results from a cross-sectional study
AUTHORS	Hossain, Md. Shabab; Das, Subhasish; Gazi, Md. Amran; Alam, Md. Ashraf; Haque, Nur Muhammad Shahedul; Mahfuz, Mustafa; Ahmed, Tahmeed; Damman, Chris J.

VERSION 1 – REVIEW

REVIEWER	Reviewer name: Peter Flom Institution and Country: Peter Flom Consulting USA Competing interests: None
REVIEW RETURNED	13-Jul-2019

GENERAL COMMENTS	Usually, I confine my remarks to statistical aspects of a paper, but here I have some more general comments as well. This paper may provide useful information to specialists in child growth - I am not at all expert in this field - but it needs some substantial revision to do that. A couple general comments: 1. Isn't microbiota plural?2. The authors mention linear growth a lot but a) Children don't grow linearly, growth speeds up and slows down and b) They never discuss linear growth, they only discuss length for age. A child could be very low on LAZ yet be growing at a normal rate. Perhaps the authors mean "length" rather than "linear growth".3. Throughout "multivariate" should be "multiple" and "bivariate" should be "simple". (Multivariate models have several DV).4. LAZ is clearly very skewed and the relationship with fecal PH looks heteroscedastic. Therefore linear regression is probably not appropriate (the authors did not show any plots of residuals, but it seems likely they are not normally distributed). Quantile regression could be used.5. What database was LAZ based on?6. Was colinearity investigated? It seems likely to be present7. p values should not be listed as 0.00. The editors can recommend whether to use < 0.01 or some other format. Peter Flom p 4 line 7 What effect? The authors have not identified one. Do they mean that malnutrition is more common? That there are more deaths? That malnutrition causes more deaths? Or what?
--

	line 14 Among whom? Southern Asia compared to what other region? Maximum number or proportion? If "Southern Asia" means something like "India, Pakistan and Bangladesh" then of course it has a huge number. That's almost 2 billion people (and a high proportion of them are children). line 16 what is reduced neuro-developmental potential? and what is full developmental potential? How is it measured? Reduced and maximum compared to what? Line 31 No cycle has been identified. p. 12 line 12 What is "optimum" growth? line 16 Fecal pH cannot possibly be a determinant of growth. What the authors are positing is that it is an indicator of poor gut microbiota and that microbiota are a determinant of growth.
--	---

REVIEWER	Reviewer name: Lee E Voth-Gaeddert Institution and Country: Environmental Health Engineer, Tufts University, United States Competing interests: None
REVIEW RETURNED	14-Jul-2019

GENERAL COMMENTS	Congratulations on an interesting manuscript which attempts to demonstrate the association of fecal pH to child height for age. This provides the field with another potential cheap tool to assess changes in the gut microbiome associated with risk factors of child stunting. Overall, I think this manuscript is a good fit for BMJ Pediatrics, however, I would suggest some minor revisions. I attempt to be as objective as possible, please feel free to push back where you feel I am not. Again, well done! Overall There are a number of grammatical errors throughout the manuscript. I would suggest having someone reread it and make changes to address these issues. A suggestion (take it or leave it); you use the phrase, "fecal pH has emerged as a possible determinant of linear growth" multiple times in the manuscript (abstract and discussion). This may be a bit confusing to readers, did it emerge because of a long set of publications have showed its useful? Did it emerge because of your study? I would suggest being targeted with your statement here, especially since it is a concluding statement. Perhaps exchanging it for one sentence on what needs to happen now/next to get to the point where any new study can feel comfortable in adding this to their protocol. SIBO is given its own paragraph in the methods section, yet this is the only place in the text it is talked about. I think it is important to note it was not correlated. Please add this to the intro, results and discussion. If not, remove it from the manuscript. Abstract
--

Pg 3; lines 17-23: Since BEED is a longitudinal study, please state that this is a cross-sectional study. Assuming those two things are correct.

Pg 4; line 5: malnutrition does not directly cause 50% of all child deaths, it either contributes to or is an underlying cause of 50% of all child deaths.

Pg 4; lines 50-54: as dysbiosis is a difficult term still, I might suggest removing, "... which refers to reduction or absence of a species or presence of an inflammogenic species..." and just leave the term dysbiosis to mean "...an altered microbiota composition that could lead to a number of disease states..." Potentially to many other interactions you would need to list that could be classified as "dysbiosis" (overgrowth, opportunistic pathogens, etc).

Pg 5; lines 11-24: I think attempting to state the exact state of evidence is important in any paper. You do correctly state that generally the evidence between stunting and the microbiome are lacking, however, I think (my opinion only) there is more evidence for bifido than clostridia. Even our recent stunting-microbiome paper from Guatemala (<https://doi.org/10.1089/ees.2019.0104>) suggested bifido was more prevalent, but we found mixed results for clostridia. I might suggest ordering them accordingly and possible stating that fact (if you agree with my statement) of the state of the evidence to date (i.e. bifido has medium evidence and clostridia has less, but both have biological plausibility).

Pg 6; lines 4-20: You provide the citation for the BEED study, but even the BEED study paper does not have all references for protocols listed here. Please add citations to protocols used. 1) Weight and length, 2) validation of questionnaire, and 3) FFQ.

Pg 6; lines 33-44: why is reference 19 cited here? Reference 21, as that describes how you collected feces. Since it is buried in the supplementary file of reference 21, please be a bit more specific on fecal sample collection. How did you collect the fecal samples (i.e. sanitary diaper)? Did you get them all in one place and wait for them to defecate? Was it the first bowel movement after enrollment (i.e. pre-nutrition intervention)?

Pg 6; 48-57; this is the first and last time you talk about SIBO. I think it is interesting to know nothing was found associated with SIBO. If you keep SIBO in, you need to add a sentence or two in the intro, results and discussion. If not, please remove. But again, I think it is good to know this was not associated. Furthermore, please provide a reference to the protocol for your application of the BreathTracker SC.

Also, were samples from adults only taken from a subset of them (i.e. malnourished ones)? How many? Why did you do this for adults and not the children? Please clarify in the manuscript.

Pg 7; line 16; would suggest changing "... using Pearson correlation and scatter plots." to "... using Pearson correlation and visualized by scatter plots." Or something to that effect.

Pg 7; line 49; please include these results (pH and LAZ) in the results table (i.e. Table 1). If you have WAZ and WHZ please report those as well (at least in the Table 1).

	This provides the reader context of the child's health. A rule of thumb I follow is that I should be able to look at the title and tables/figures and understand what the study was about and what the primary outcomes were. Currently, I have to go digging in the text. Pg 8; line 4; I am not sure which variables were run through the bivariate analysis and which ones made it to the next round. Please be explicit. Is the 'unadjusted' columns in Table 2 all the variables you ran in the bivariate analysis (assuming that is why you called it unadjusted)? State that in the text so the reader knows what you looked for. What were the variables that made it to the next round? Did only WAMI and mother's height make it? Is that what the 'adjusted' column is? I might list these two in the text if it is only two additional variables. Lastly, I would prefer if you use the terms 'associated with' or 'correlated with' (using 'resulted in' or 'caused' infers you tested for causality, which you did not, just correlation). Pg 8; line 22; remove the second sentence of the paragraph. Your paper is unique, that is why you are publishing it. Pg 9; lines 31-33; you write your sentence as if 12 months of age is the recommended time to do complimentary feeding along with breastfeeding. This is not what the WHO recommends (at least not the last time I checked). WHO states breastfeeding should ideally continue up to 2 years of age or longer. If you are citing a different source, please make this reference. Pg 10; line 3: I may have missed it but was this acronym defined previously? Pg 12; line 18: if possible (with word count), may reiterate the limited state of evidence we currently have of this (i.e. match how you describe it in the intro). Pg 16; line 8: if you use B for regression co-efficient, I would suggest using the Beta symbol. Your third "sig" column should be p-value (you use p-value in the text, which is the correct term). Again, well done and congrats. Happy to clarify any statements made.
--	---

VERSION 1 – AUTHOR RESPONSE

Reviewer 1

1. Isn't microbiota plural?

Response: Thank you very much for your concern. The noun microbiota can be countable or uncountable. There are mixed opinions about the word being singular or plural. In some references it has been stated as singular while in others plural. In general contexts, both the singular and plural form can be microbiota. However, in more specific contexts, the plural form can also be microbiotas e.g. in reference to various types of microbiotas or a collection of microbiotas.

2. The authors mention linear growth a lot but a) Children don't grow linearly, growth speeds up and slows down and b) They never discuss linear growth, they only discuss length for age. A child could be very low on LAZ yet be growing at a normal rate. Perhaps the authors mean "length" rather than "linear growth".

Response: Thank you very much for your concern. Yes, length was measured only at a single time point in this study, so it will not be linear growth, rather it would be length. We have replaced the term linear growth and revised accordingly throughout the manuscript (Page 1,4,6,9, 12, 13, 14).

3. Throughout "multivariate" should be "multiple" and "bivariate" should be "simple". (Multivariate models have several DV).

Response: Thank you very much for your concern. We have revised accordingly (Page 4, 8, 9).

4. LAZ is clearly very skewed and the relationship with fecal PH looks heteroscedastic. Therefore linear regression is probably not appropriate (the authors did not show any plots of residuals, but it seems likely they are not normally distributed). Quantile regression could be used.

Response: Thank you very much for your concern. Yes, the LAZ is not normally distributed. We performed the Shapiro-Wilk test for testing normality and the p-value was <0.01 . So as per your valuable suggestion, we performed quantile regression and found the result to be similar in terms of its association with fecal pH. The co-efficient value did not vary by much either. However, the previous positive association between LAZ and maternal height was not found in this case. We have revised the manuscript according to the new results (Page 4, 8, 9, 19, 20).

5. What database was LAZ based on?

Response: Thank you very much for your concern. LAZ was based on the WHO 2006 database.

6. Was collinearity investigated? It seems likely to be present.

Response: Thank you very much for your concern. Yes, collinearity was investigated and it was absent. The threshold for VIF was considered as 3. And the highest VIF result was 1.39 for LAZ with MDD. So, no multicollinearity issues were present.

7. p values should not be listed as 0.00. The editors can recommend whether to use < 0.01 or some other format.

Response: Thank you very much for your concern. We have revised the format in the results section accordingly (Page 4, 9, 19, 20).

8. P 4 line 7 What effect? The authors have not identified one. Do they mean that malnutrition is more common? That there are more deaths? That malnutrition causes more deaths? Or what?

Response: Thank you very much for your concern. We meant that malnutrition is more common in low-and-middle-income countries (LMICs). We have revised and rewritten the statement accordingly (Page 5).

9. Line 14 Among whom? Southern Asia compared to what other region? Maximum number or proportion? If "Southern Asia" means something like "India, Pakistan and Bangladesh" then of course it has a huge number. That's almost 2 billion people (and a high proportion of them are children).

Response: Thank you very much for your concern. By Southern Asia or South Asia we meant the southern region of the Asian continent, which comprises the sub-Himalayan countries, more specifically, Bangladesh, Bhutan, India, Nepal and Pakistan.

In comparison to other developing countries across the world, the stunting rate of children in this region is substantially high. We have revised and rewritten the statement accordingly (Page 5).

10. Line 16 what is reduced neuro-developmental potential? and what is full developmental potential? How is it measured? Reduced and maximum compared to what?

Response: Thank you very much for your concern. By neuro-developmental potential we meant the potential chance of a proper development of neurological and cognitive function. Studies show that stunting has a negative impact on neurological development and cognitive function in pre-school children. This results in sub-standard educational performance and economic productivity in their later phases of life. Reduced developmental potential refers to delayed cognition, behavioural problems and later poor school achievement that persist till adulthood (Grantham-McGregor, 2007). The children's development (DQ) is measured by the Griffiths mental development scales. Stunted children suffer from delayed cognition and behavioural problems, which tremendously affects their productivity in the later phases of life in, comparison to their non-malnourished counterparts.

11. Line 31 No cycle has been identified.

Response: Thank you very much for your concern. We have revised and rewritten the statements accordingly (Page 5).

12. P. 12 line 12 What is "optimum" growth?

Response: Thank you very much for your concern. By optimum linear growth we referred to children having a normal length for age Z-score based on the WHO Child Growth Standards.

13. Line 16 Fecal pH cannot possibly be a determinant of growth. What the authors are positing is that it is an indicator of poor gut microbiota and that microbiota are a determinant of growth.

Response: Thank you very much for your concern. Yes, fecal pH is an indicator of gut microbiota status and the microbiota are in turn a determinant of growth. We have revised and rewritten the statements accordingly (Page 4, 12, 13).

Reviewer 2

1. A suggestion (take it or leave it); you use the phrase, "fecal pH has emerged as a possible determinant of linear growth" multiple times in the manuscript (abstract and discussion). This may be a bit confusing to readers, did it emerge because of a long set of publications have showed its useful? Did it emerge because of your study? I would suggest being targeted with your statement here, especially since it is a concluding statement. Perhaps exchanging it for one sentence on what needs to happen now/next to get to the point where any new study can feel comfortable in adding this to their protocol.

Response: Thank you very much for your concern. Yes, fecal pH is an indicator of gut microbiota status and the microbiota are in turn a determinant of growth. In this study, the association between fecal pH and growth is indirect and is based on the gut microbiota status. Characterization of microbiome (SCFA producers) in the stool could not be investigated in the present study, which is the only way to clarify this topic and remained a limitation. Further studies are warranted to make exploration in this regard. We have revised and rewritten the statements accordingly (Page 4, 12, 13).

2. SIBO is given its own paragraph in the methods section, yet this is the only place in the text it is talked about. I think it is important to note it was not correlated. Please add this to the intro, results and discussion. If not, remove it from the manuscript.

Response: Thank you very much for your concern. We have added SIBO in the intro, results and discussion and revised the manuscript accordingly (Page 5, 9, 11).

3. Pg 3; lines 17-23: Since BEED is a longitudinal study, please state that this is a cross-sectional study. Assuming those two things are correct.

Response: Thank you very much for your concern. Yes, this is a cross-sectional study. We have revised and rewritten the statement accordingly (Page 4, 5, 7).

4. Pg 4; line 5: Malnutrition does not directly cause 50% of all child deaths, it either contributes to or is an underlying cause of 50% of all child deaths.

Response: Thank you very much for your concern. Yes, malnutrition does not directly cause but acts as an underlying cause of these child deaths. We have revised and rewritten the statement accordingly (Page 5).

5. Pg 4; lines 50-54: as dysbiosis is a difficult term still, I might suggest removing, "... which refers to reduction or absence of a species or presence of an inflammogenic species..." and just leave the term dysbiosis to mean "...an altered microbiota composition that could lead to a number of disease states..." Potentially to many other interactions you would need to list that could be classified as "dysbiosis" (overgrowth, opportunistic pathogens, etc).

Response: Thank you very much for your concern. We have revised and rewritten the statement accordingly (Page 6).

6. Pg 5; lines 11-24: I think attempting to state the exact state of evidence is important in any paper. You do correctly state that generally the evidence between stunting and the microbiome are lacking, however, I think (my opinion only) there is more evidence for bifido than clostridia. Even our recent stunting-microbiome paper from Guatemala (<https://doi.org/10.1089/ees.2019.0104>) suggested bifido was more prevalent, but we found mixed results for clostridia. I might suggest ordering them accordingly and possibly stating that fact (if you agree with my statement) of the state of the evidence to date (i.e. bifido has medium evidence and clostridia has less, but both have biological plausibility).

Response: Thank you very much for your concern and reference. Yes, we agree with your statement. We have ordered them accordingly, included this important reference and revised the statements in the manuscript (Page 4, 6, 10, 11).

7. Pg 6; lines 4-20: You provide the citation for the BEED study, but even the BEED study paper does not have all references for protocols listed here. Please add citations to protocols used. 1) Weight and length, 2) validation of questionnaire, and 3) FFQ.

Response: Thank you very much for your concern. As mentioned in the 'Monitoring and Quality Control Measures' section of the BEED study manuscript, previously established and validated SOPs from the MAL-ED study were used in this study as well. For weight and length, reference from anthropometric measurements in the WHO Multicentre Growth Reference Study was used. We have added the citations and revised accordingly (Page 7)

8. Pg 6; lines 33-44: why is reference 19 cited here? Reference 21, as that describes how you collected feces. Since it is buried in the supplementary file of reference 21, please be a bit more specific on fecal sample collection. How did you collect the fecal samples (i.e. sanitary diaper)?

Did you get them all in one place and wait for them to defecate? Was it the first bowel movement after enrollment (i.e. pre-nutrition intervention)?

Response: Thank you very much for your concern. Prior to the nutritional intervention, the health workers waited for the children to defecate and faecal samples from the first bowel movement after enrolment were collected into a plastic pot and 1-2 grams from that sample was relocated to a sterile specimen container. We have revised and rewritten the section accordingly (Page 7, 8).

9. Pg 6; 48-57; this is the first and last time you talk about SIBO. I think it is interesting to know nothing was found associated with SIBO. If you keep SIBO in, you need to add a sentence or two in the intro, results and discussion. If not, please remove. But again, I think it is good to know this was not associated. Furthermore, please provide a reference to the protocol for your application of the BreathTracker SC. Also, were samples from adults only taken from a subset of them (i.e. malnourished ones)? How many? Why did you do this for adults and not the children? Please clarify in the manuscript.

Response: Thank you very much for your concern. We have added SIBO in the intro, results and discussion and revised the manuscript accordingly (Page 5, 9, 11). The reference to the protocol for the application of the BreathTracker SC has also been added (Page 8). Samples were taken both from the adults and children, and as this manuscript deals with child participants, we included their results only. The previous phrase 'malnourished adults' was a typo, and we have revised accordingly (Page 5, 8, 9, 11).

10. Pg 7; line 16; would suggest changing "... using Pearson correlation and scatter plots." to "... using Pearson correlation and visualized by scatter plots." Or something to that effect.

Response: Thank you very much for your concern. We have revised and rewritten accordingly (Page 8).

11. Pg 7; line 49; please include these results (pH and LAZ) in the results table (i.e. Table 1). If you have WAZ and WHZ please report those as well (at least in the Table 1). This provides the reader context of the child's health. A rule of thumb I follow is that I should be able to look at the title and tables/figures and understand what the study was about and what the primary outcomes were. Currently, I have to go digging in the text.

Response: Thank you very much for your concern. We have included all the results in Table 1 accordingly (Page 18).

12. Pg 8; line 4; I am not sure which variables were run through the bivariate analysis and which ones made it to the next round. Please be explicit. Is the 'unadjusted' columns in Table 2 all the variables you ran in the bivariate analysis (assuming that is why you called it unadjusted)? State that in the text so the reader knows what you looked for. What were the variables that made it to the next round? Did only WAMI and mother's height make it? Is that what the 'adjusted' column is? I might list these two in the text if it is only two additional variables. Lastly, I would prefer if you use the terms 'associated with' or 'correlated with' (using 'resulted in' or 'caused' infers you tested for causality, which you did not, just correlation).

Response: Thank you very much for your concern. Yes, all the variables in table 2 were run through the bivariate analysis, hence termed unadjusted. As LAZ is the outcome variable, at first bivariate quantile regression was performed between LAZ scores with each individual factor. Results with a significance level at or below 0.2 were included in the multivariate regression model. In addition, age and sex were included in the final model due to the inconsistent growth pattern in children of either sex in this age group as recommended in certain studies and minimum dietary diversity (MDD) was included in the final model instead of the individual food groups.

Multivariate quantile regression was done to quantify the relation between fecal pH and LAZ scores after adjusting for the variables and finally age, sex, maternal height, and MDD were included in the model, hence the column is termed as adjusted. We have explained this in text in the 'Statistical analysis' section (Page 8). We have also revised the terms accordingly (Page 9).

13. Pg 8; line 22; remove the second sentence of the paragraph. Your paper is unique, that is why you are publishing it.

Response: Thank you very much for your concern. We have removed the sentence from the paragraph accordingly (Page 9).

14. Pg 9; lines 31-33; you write your sentence as if 12 months of age is the recommended time to do complimentary feeding along with breastfeeding. This is not what the WHO recommends (at least not the last time I checked). WHO states breastfeeding should ideally continue up to 2 years of age or longer. If you are citing a different source, please make this reference.

Response: Thank you very much for your concern. Yes, as per WHO recommendation breastfeeding should ideally continue up to 2 years of age. We have revised and rewritten accordingly (Page 11).

15. Pg 10; line 3: I may have missed it but was this acronym defined previously?

Response: Thank you very much for your concern. Yes, the acronym has been previously defined in the second paragraph of the 'Introduction' section (Page 5).

16. Pg 16; line 8: if you use B for regression co-efficient, I would suggest using the Beta symbol. Your third "sig" column should be p-value (you use p-value in the text, which is the correct term).

Response: Thank you very much for your concern. We have revised and rewritten accordingly (Page 19, 20).